# DOES DEEP ACTIVE LEARNING WORK IN THE WILD?

## ABSTRACT

Deep active learning (DAL) methods have shown significant improvements in sample efficiency compared to simple random sampling. While these studies are valuable, they nearly always assume that optimal DAL hyperparameter (HP) settings are known in advance, or optimize the HPs through repeating DAL several times with different HP settings. Here, we argue that in real-world settings, or *in the wild*, there is significant uncertainty regarding good HPs, and their optimization contradicts the premise of using DAL (i.e., we require labeling efficiency). In this study, we evaluate the performance of eleven modern DAL methods on eight benchmark problems as we vary a key HP shared by all methods: the pool ratio. Despite adjusting only one HP, our results indicate that eight of the eleven DAL methods sometimes underperform relative to simple random sampling and some frequently perform worse. Only three methods always outperform random sampling (albeit narrowly), and we find that these methods all utilize diversity to select samples - a relatively simple criterion. Our findings reveal the limitations of existing DAL methods when deployed *in the wild*, and present this as an important new open problem in the field.

## 1 INTRODUCTION

In this work, we focus on the application of active learning to deep neural networks (DNNs), sometimes referred to as Deep Active Learning (DAL) Roy et al. (2018). Broadly speaking, the premise of DAL is that some training instances will yield superior performance compared to others. Therefore, we can improve the training sample efficiency of DNNs by selecting the best training instances. A large number of methods have been investigated in recent years for DAL Settles (2009); Ren et al. (2021); Holzmüller et al. (2023), often reporting significant improvements in sample efficiency compared to simpler strategies, such as random sampling Tsymbalov et al. (2018); Käding et al. (2018); Kee et al. (2018). While these studies provide valuable insights, they nearly always assume good DAL hyperparameter (HP) settings are known in advance, or alternatively, they optimize the HPs (e.g., by repeating DAL several times with different HP settings). To our knowledge however, there is little evidence that one can assume good hyperparameters are known in advance for novel problems (see Section 3, where we find HP settings in the literature vary widely across problems). Moreover, running a DAL method multiple times in search of good HP settings may result in significant label inefficiency, even when compared to random sampling. Therefore, in real-world settings where DAL is applied to a novel problem, or *in the wild* as we term it here, the best DAL HPs are not generally known in advance, and it is unclear whether DAL still offers advantages (e.g., compared to random sampling) when accounting for HP uncertainty. If DAL models do not reliably outperform simple random sampling in the presence of HP uncertainty, it greatly undermines their value, and the likelihood that they will be adopted. Despite the significance of this problem, it has received little attention in the literature.

**Contributions** In this work, we perform the first systematic evaluation of DAL *in the wild*. We focus our investigation on DAL for regression, where to our knowledge, most applicable DAL methods are pool-based, and therefore they share an important HP: the pool ratio, $\gamma$ (see Section 3.3). Using this property of regression problems, we evaluate a large number of DAL models as we vary a single HP, their $\gamma$ setting, thereby providing a *distribution* of performance that one can expect in real-world settings (i.e., in the wild), where the best setting for $\gamma$ is uncertain. We note that most DAL models have several (often unique) HPs that exhibit uncertainty, and each can contribute to performance variability of DAL methods in the wild. However, examining variability with respect to all of these

Figure 1: Schematic diagram for pool-based DAL procedure. In the input space X, the triangles represent labeled data ($L$), and the circles represent unlabeled data ($D$ for the full set of unlabeled data, and $U$ for subsampled unlabeled pool). At each step, after the model is trained using the existing training set $L$, a subset of unlabeled data $U$ is sampled and evaluated by the AL criteria q(x). Then, the top-k points according to q(x) are labeled by the oracle function.

HPs would require a lengthy exposition, and would be computationally costly. Therefore we focus on $\gamma$, which mitigates the aforementioned challenges, while still providing sufficient empirical evidence to support our main conclusions.

To support our investigation, we assembled eight scientific computing regression problems to examine the performance of DAL methods in this setting; to our knowledge, this is the first such benchmark of its kind. We then identified past and recent DAL methods that are suitable for regression, totaling eleven methods

To support our study, we identified eleven DAL methods that are suitable for regression. We then examined the performance of these DAL methods on each of eight benchmark problems, compared to simple random sampling, as we vary their $\gamma$ settings. Our results indicate that their performance varies significantly with respect to $\gamma$, and that the best HP varies for different DAL/dataset with no single $\gamma$ value working best across all settings, confirming our hypothesis that there is significant uncertainty regarding the best HP setting for novel problems. We also find that most of the DAL methods sometimes underperform simple random sampling and some frequently perform much worse:

- We compile a large benchmark of eleven state-of-the-art DAL methods across eight datasets. For some of our DAL methods, we are the first to adapt them to regression. Upon publication, we will publish the datasets and code to facilitate reproducibility.

- Using our benchmark, we perform the first analysis of DAL performance *in the wild*. Using $\gamma$ as an example, we systematically demonstrate the rarely-discussed problem that most DAL models are often outperformed by simple random sampling when we account for HP uncertainty.

- We analyze the factors that contribute to the robustness of DAL in the wild, with respect to $\gamma$.

## 2 RELATED WORKS

### 2.1 ACTIVE LEARNING BENCHMARKS

The majority of existing AL benchmarks are for classification tasks, rather than regression Jose et al. (2024), and many AL methods for classification cannot be applied to regression. Some existing studies include Zhan et al. (2021), which benchmarked AL using a Support Vector Machine (SVM) with 17 AL methods on 35 datasets. Yang & Loog (2018) benchmarked logistic regression with 12 AL methods and 44 datasets. Meduri et al. (2020) benchmarked specific entity matching application (classification) of AL with 3 AL methods on 12 datasets, with 3 different types of classifiers (DNN, SVM, and Tree-based). Trittenbach et al. (2021) benchmarked an AL application in outlier detection on 20 datasets and discussed the limitation of simple metrics extensively. Hu et al. (2021) benchmarked 5 classification tasks (including both image and text) using DNN. Beck et al. (2021) benchmarked multiple facets of DAL on 5 image classification tasks. For the regression AL

benchmark, O'Neill et al. (2017) benchmarked 5 AL methods and 7 UCI [1] datasets, but they only employed linear models. Wu et al. (2019) compared 5 AL methods on 12 UCI regression datasets, also using linear regression models. Our work is fundamentally different from both, as we use DNNs as our regressors, and we employ several recently-published problems that also involved DNN regressors, making them especially relevant for DAL study. The recent study by Holzmüller et al. (2023) is the only work that is similar to ours, in which the authors benchmarked 8 pool-based DAL methods for regression on 15 datasets. The primary focus of their work was to propose a novel DAL regression framework, termed LCMD; meanwhile the focus of our work is to investigate DAL in the wild. Consequently, Holzmüller et al. (2023) presents different performance metrics and conclusions compared to our study.

## 2.2 ACTIVE LEARNING FOR REGRESSION PROBLEMS

Regression problems have received (relatively) little attention compared to classification Ren et al. (2021); Guyon et al. (2011). For the limited AL literature dedicated to regression tasks, Expected Model Change (EMC) Settles (2008); Cai et al. (2013) was explored, where an ensemble of models was used to estimate the true label of a new query point using both linear regression and tree-based regressors. Gaussian processes were also used with a natural variance estimate on unlabeled points in a similar paradigm Käding et al. (2018). Smith et al. (2018) used Query By Committee (QBC), which trains multiple networks and finds the most disagreeing unlabeled points of the committee of models trained. Tsymbalov et al. (2018) used the Monte Carlo drop-out under a Bayesian setting, also aiming for maximally disagreed points. Yu & Kim (2010) found $x$-space-only methods outperforming y-space methods in robustness. Yoo & Kweon (2019) proposed an uncertainty-based mechanism that learns to predict the loss using an auxiliary model that can be used on regression tasks. Ranganathan et al. (2020) and Käding et al. (2016) used Expected Model Output Change (EMOC) with Convolutional Neural Network (CNN) on image regression tasks with different assumptions. We included all these methods that used deep learning in our benchmark.

## 2.3 DAL IN THE WILD

To our knowledge, all empirical studies of pool-based DAL methods assume that an effective pool ratio hyperparameter, $\gamma$, is known apriori. While the majority of works assumed the original training set as the fixed, unlabeled pool, Yoo & Kweon (2019) limited their method to a subset of 10k instances instead of the full unlabeled set and Beluch et al. (2018) used subsampling to create the pool $U$ (and hence $\gamma$). In real-world settings - in the wild - we are not aware of any method to set $\gamma$ a priori, and there has been no study of DAL methods under this setting. Therefore, we believe ours is the first such study.

## 3 PROBLEM SETTING

In this work, we focus on DAL for regression problems, which comprise a significant portion of DAL problems involving DNNs Jose et al. (2024). As discussed in Section 1, nearly all DAL methods for regression are pool-based, which is one of the three major paradigms of AL, along with stream-based and query synthesis. Settles (2009)

## 3.1 FORMAL DESCRIPTION

Let $L^i = (X^i, Y^i)$ be the dataset used to train a regression model at the $i^{th}$ iteration of active learning. We assume access to some oracle, denoted $f : \mathcal{X} \rightarrow \mathcal{Y}$, that can accurately produce the target values, $y \in \mathcal{Y}$ associated with input values $x \in \mathcal{X}$. Since we focus on DAL, we assume a DNN as our regression model, denoted $\hat{f}$. We assume that some relatively small number of $N_0$ labeled training instances are available to initially train $\hat{f}$, denoted $L^0$. In each iteration of DAL, we must choose $k$ query instances to be labeled by the oracle, yielding a set of labeled instances, denoted $Q$, that is added to the training dataset. Our goal is then to choose $Q$ that maximizes the performance of the DNN-based regression models over unseen test data at each iteration of active learning.

---

[1] University of California Irvine Machine Learning Repository

## 3.2 POOL-BASED DEEP ACTIVE LEARNING

General pool-based DAL methods assume that we have some pool $U$ of $N_U$ unlabeled instances from which we can choose the $k$ instances to label. The set $U$ is sampled from a larger and potentially-infinite set, denoted $D$, and $N_U$ is a HP chosen by the DAL user. We note that in some DAL applications, such as computer vision, it is conventional to utilize all available unlabeled data for $U$, and the pool size is not often explicitly varied or discussed. However, this convention is equivalent to setting $U = D$, and thereby implicitly setting the $N_U$ HP. Most pool-based methods rely upon some acquisition function $q : \mathcal{X} \to \mathbb{R}$ to assign some scalar value to each $x \in U$ indicating its "informativeness", or utility for training $\hat{f}$. In each iteration of active learning, $q$ is used to evaluate all instances in $U$, and the top $k$ are chosen to be labeled and included in $L$.

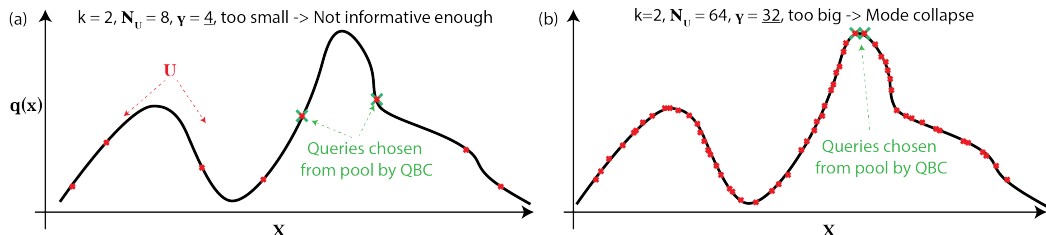

Figure 2: Pool-based DAL for uncertainty-based mechanism. $q(x)$ is the acquisition metric. (a, b) are two scenarios of the pool ratio ($\gamma$) being too small (4 in a) or too large (32 in b) in $k$ (step size) of 2.

## 3.3 THE POOL RATIO HYPERPARAMETER, $\gamma$

We define the *pool ratio* as $\gamma = N_U/k$. By definition, $N_U$ and $k$ are hyperparameters of pool-based problems, and therefore $\gamma$ also is. While one could, in principle, vary $N_U$ and $k$ independently, this is not often done in practice. Typically $k$ is set as small as possible, limited by computational resources. This leaves $N_U$ as the major free hyperparameter; however, prior research has found that its impact depends strongly on its size relative to $k$ Kee et al. (2018); Tsymbalov et al. (2018); Käding et al. (2018), encoded in $\gamma$. Given a fixed value of $k$, increasing $N_U$ can lead to the discovery of points with higher values of $q(x)$ due to denser sampling of the input space. However, a larger $N_U$ also increases the similarity of the points, which provides redundant information to the model - a problem referred to as mode collapse Burbidge et al. (2007); Ren et al. (2021); Kee et al. (2018). In the limit as $N_U \to \infty$, all of the $k$ selected query points will be located near the same $x \in \mathcal{X}$ that has the highest value of $q(x)$. This tradeoff is illustrated in Fig. 2 for a simple problem, and has also been noted in Cacciarelli & Kulahci (2024).

In most real-world settings, there is a substantial quantity of unlabeled data (often infinite), and the user has the freedom (or burden) of choosing a suitable $\gamma$ setting for their problem by varying the size of $U$. Crucially, and as we show in our experiments, choosing a sub-optimal $\gamma$ value can result in poorer performance than naive random sampling. This is not necessarily a problem if either (i) one $\gamma$ setting works across most problems or, alternatively, (ii) $\gamma$ can be optimized on new problems without using labels. To the best of our knowledge, there is no method for optimizing $\gamma$ on a new problem without running multiple trials of AL to find the best one (i.e., collecting labels), defeating the purpose of AL in real-world settings. Furthermore, the value of $\gamma$ varies widely across the literature, suggesting that suitable settings for $\gamma$ indeed vary across problems (see supplement for a list).

## 4 BENCHMARK REGRESSION PROBLEMS

To compose our benchmarks, we focused primarily upon problems in scientific computing, which is an important emerging problem setting Subramanian et al. (2024); Takamoto et al. (2022); Majid & Tudisco (2024). We propose eight regression problems to include in our benchmark set: two simple toy problems (SINE, ROBO), four contemporary problems from publications in diverse fields of science and engineering (STACK, ADM, FOIL, HYDR) and two problems solving ordinary

Table 1: Benchmark datasets dimensionality and oracle functions. $Dim_{x,y}$ are the dimensionality of $x$ and y. Note that ODE solutions are implemented in the form of analytical functions as well.

| DATASET | SINE | ROBO | STACK | ADM | FOIL | HYDR | BESS | DAMP |
|---|---|---|---|---|---|---|---|---|
| $Dim_x$ | 1 | 4 | 5 | 14 | 5 | 6 | 2 | 3 |
| $Dim_y$ | 1 | 2 | 201 | 2000 | 1 | 1 | 1 | 100 |
| ORACLE | ANALYTICAL | | NUMERICAL SIMULATOR | DNN | RANDOM FOREST | | ODE SOLUTION | |

differential equations (also prevalent in engineering). Summary details of our benchmark problems can be found in Table 1 and Table 2.

We utilized four major selection criteria, beyond choosing scientific computing problems: (i) diversity: we sought to include a set of problems that span different disciplines (aero and fluid-dynamics, materials science), and problems that require physical experiments (e.g., FOIL, HYDRO) versus simulators (e.g., ADM); (ii) availability of labeled data: the problems we chose (unlike many high dimension ones) all had sufficiently large amount of labeled data, allowing us to easily study the impact of different pool ratios; (iii) dimensionality: we sought problems with relatively low dimensionality because they mitigate computational costs allowing for more extensive experimentation, while still being representative of many contemporary scientific computing problems (e.g., labeling can be highly expensive, severely limiting total labeled data, and making even low-dimensional problems challenging); (iv) difficulty: the problems in our dataset are also "difficult" in the sense that the accuracy of the learners (i.e., the DNN regressors) can vary significantly depending upon which data are labeled, making it possible to distinguish between more/less effective AL approaches. Although this is not the only notion of "difficulty" that may be relevant for selecting benchmark problems, we believe this is the most important one, and has been used in recent DAL studies Holzmüller et al. (2023). We now describe our benchmark problems:

Table 2: Details of used oracles. Along with details in our repository (to be made public after publication), we provide information on the oracles' source publication, type of ML model, source of training data, quantity of data, and error level. For all datasets, we adopt an 80-20 train-test split.

| DATASET | TYPE OF ML MODEL | SOURCE OF DATA | QUANTITY OF DATA | TEST MSE |
|---|---|---|---|---|
| ADM Deng et al. (2021b) | ENSEMBLE OF DNNs | NUMERICAL SIMULATOR | 160K SAMPLES | 6.00E-05 |
| FOIL Dua & Graff (2017) | RANDOM FOREST | REAL-WORLD EXPERIMENTS | 1503 SAMPLES | 8.63E-03 |
| HYDRO Dua & Graff (2017) | RANDOM FOREST | REAL-WORLD EXPERIMENTS | 302 SAMPLES | 3.97E-02 |

**1D sine wave (SINE)** A noiseless 1-dimensional sinusoid with smoothly-varying frequency. **2D robotic arm (ROBO)** Ren et al. (2020) The goal is to predict the 2-D spatial location of the endpoint of a robotic arm based on its three joint angles. **Stacked material (STACK)** Chen et al. (2019) The goal is to predict the 201-D reflection spectrum of a material based on the thickness of its five layers. **Artificial Dielectric Material (ADM)** Deng et al. (2021b) The goal is to predict the 2000-D reflection spectrum of a material based on its 14-D geometric structure. Full wave electromagnetic simulations were utilized in Deng et al. (2021a) to label data in the original work, requiring 1-2 minutes per input point. **NASA Airfoil (FOIL)** Dua & Graff (2017) The goal is to predict the sound pressure of an airfoil based on the structural properties of the foil, such as its angle of attack and chord length. This problem was published by NASA Brooks et al. (1989) and the instance labels were obtained from a series of real-world aerodynamic tests in an anechoic wind tunnel. It has been used in other AL literature Wu (2018); Liu & Wu (2020); Jose et al. (2024). **Hydrodynamics (HYDR)** Dua & Graff (2017) The goal is to predict the residual resistance of a yacht hull in water based on its shape. This problem was published by the Technical University of Delft, and the instance labels were obtained by real-world experiments using a model yacht hull in the water. It is also referred to as the "Yacht" dataset in some AL literature Wu et al. (2019); Cai et al. (2013); Jose et al. (2024). **Bessel function (BESS)** The goal is to predict the value of the solution to Bessel's differential equation, a second-order ordinary differential equation that is common in many engineering problems. The inputs are the function order $\alpha$ and input position $x$. The order $\alpha$ is limited to non-negative integers below 10. **Damping Oscillator (DAMP)** The goal is to predict the full-swing trajectory of a damped

Table 3: List of benchmarked methods. $L$ is the labeled set, $Q$ is the already selected query points with $dist$ being L2 distance, $\hat{f}(x)$ is model estimate of $x$, $f(x)$ is oracle label of $x$, $\mu(x)$ is the average of ensemble model output, N is number of models in the ensemble, $N_k$ is the k-nearest-neighbors, $sim$ is cosine similarity, $\phi$ is current model parameter, $\phi'$ is the updated parameter, $\mathcal{L}(\phi; (x', y'))$ is the loss of model with parameter $\phi$ on new labeled data $(x', y')$, $f_{loss}(x)$ is the auxiliary model that predicts the relative loss

| METHOD | ACQUISITION FUNCTION (Q) |
|---|---|
| CORE-SET (GSX) SENER & SAVARESE (2017) | $\min\limits_{x\in\mathcal{L}\cup\mathcal{Q}} dist(x^*, x)$ |
| GREEDY SAMPLING IN Y (GSY) WU ET AL. (2019) | $\min\limits_{y\in\mathcal{L}\cup\mathcal{Q}} dist(\hat{f}(x^*), y)$ |
| IMPROVED GREEDY SAMPLING (GSXY) WU ET AL. (2019) | $\min\limits_{(x,y)\in\mathcal{L}\cup\mathcal{Q}} dist(x^*, x) * dist(\hat{f}(x^*), y)$ |
| QUERY BY COMMITTEE (QBC) KEE ET AL. (2018) | $\dfrac{1}{N}\sum\limits_{n=1}^{N}(\hat{f}_n(x^*) - \mu(x^*))^2$ |
| QBC W/ DIVERSITY (QBCDIV) KEE ET AL. (2018) | $q_{QBC}(x^*) + q_{div}(x^*)$ $(q_{div}(x^*) = q_{GSx}(x^*))$ |
| QBC W/ DIVERSITY & DENSITY (QBCDIVDEN) KEE ET AL. (2018) | $q_{QBC}(x^*) + q_{div}(x^*) + q_{den}(x^*)$ $(q_{den}(x^*) = \dfrac{1}{k}\sum\limits_{x\in N_k(x^*)} sim(x^*, x))$ |
| BAYESIAN BY DISAGREEMENT (BALD) TSYMBALOV ET AL. (2018) | $q_{QBC}(x^*)$ WITH DROPOUT |
| EXPECTED MODEL OUTPUT CHANGE (EMOC) RANGANATHAN ET AL. (2020) | $\mathbb{E}_{y'\|x'}\mathbb{E}_x\|\|\hat{f}(x^*; \phi') - \hat{f}(x^*; \phi)\|\|_1$ $\approx \mathbb{E}_x\|\|\nabla_\phi \hat{f}(x; \phi) * \nabla_\phi\mathcal{L}(\phi; (x^{*'}, y'))\|\|_1$ |
| LEARNING LOSS YOO & KWEON (2019) | $f_{loss}(x^*)$ |
| CLUSTER-VARIANCE CITOVSKY ET AL. (2021) | $q_{QBC}(x)^*$ IN CLUSTERS |
| DENSITY-AWARE CORE-SET (DACS)KIM & SHIN (2022) | $q_{GSx}(x^*) + q_{den}(x^*)$ |

oscillator in the first 100 time steps, of the solution to a second-order ordinary differential equation. The input is the magnitude, damping coefficient, and frequency of the oscillation.

From the literature, we found eleven AL methods that are (i) applicable to regression problems, (ii) with DNN-based regressors, making them suitable for benchmark regression problems. Due to space constraints, we list each method in Table 3 along with key details, and refer readers to the supplement for full details. Some of the methods have unique HPs that must be set by the user. In these cases, we adopt HP settings suggested by the methods' authors, shown in Table 3. Upon publication, we will publish code for all of these methods to support future benchmarking.

## 5 BENCHMARK EXPERIMENT DESIGN

In our experiments, we compare eleven state-of-the-art DAL methods on eight scientific computing problems. We evaluate the performance of our DAL methods as a function of $\gamma$ on each of our benchmark problems, with $\gamma \in [2, 4, 8, 16, 32, 64]$ (i.e., at each step we sample our U with $k * \gamma$ points). Following convention Kee et al. (2018); Tsymbalov et al. (2018), we assume a small training dataset is available at the outset of active learning, $T^0$, which has $N_0 = 80$ randomly sampled training instances. We then run each DAL model to $T^{50}$ AL steps, each step identifying $k = 40$ points to be labeled from a fresh, randomly generated pool of size $k * \gamma$. For each benchmark problem, we assume an appropriate neural network architecture is known apriori. Each experiment (i.e., the combination of dataset, DAL model, and $\gamma$ value) is run 5 times to account for randomness. The MSE is calculated over a set of 4000 test points that are uniformly sampled within the $x$-space boundary. To reduce unnecessary noise related to our core hypothesis, we use the same (randomly sampled) unlabeled pools across different DAL methods.

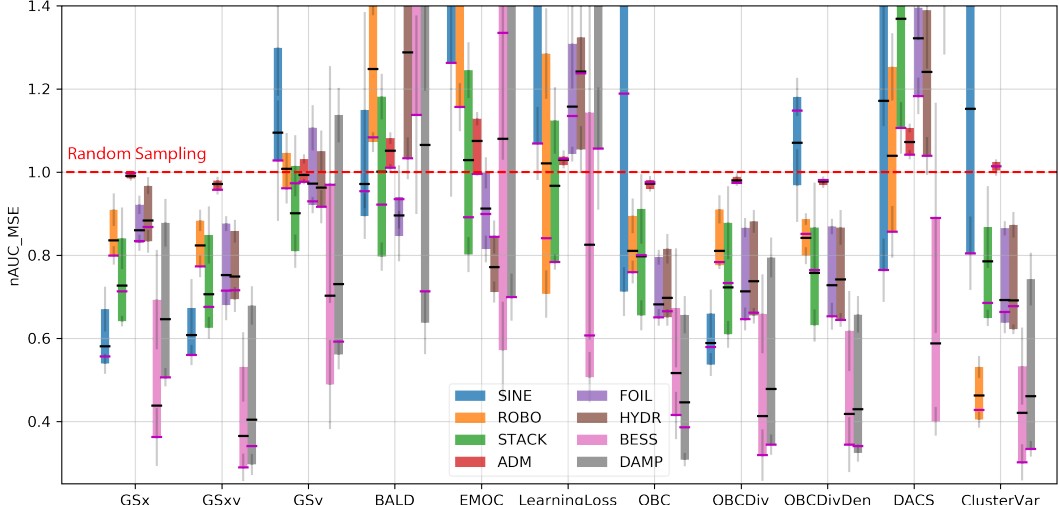

Figure 3: Performance of each DAL method (x-axis) in terms of $nAUC_{MSE}$ (y-axis). For each DAL method, we report a bar indicating the *range* of $nAUC_{MSE}$ values obtained as we vary the pool ratio, $\gamma \in [2, 4, ..., 64]$; for a given DAL method, we report one bar for each of the eight benchmark problems, indicated by a unique color in the legend. Each bar is bisected by a solid black and magenta line, respectively. The black line represents the average $nAUC_{MSE}$ value across all settings of $\gamma$. The magenta line represents the performance using $\gamma_{prior}$ (see Section 6 for details). The dashed red line at $nAUC_{MSE} = 1$ corresponds to the performance obtained using random sampling. Note that some vertical bars are intentionally clipped at the top to improve the visualization overall.

We must train a regression model for each combination of problem and DAL method. Because some DAL methods require an ensemble model (e.g., QBC), we use an ensemble of 10 DNNs as the regressor for all of our DAL algorithms (except for the ADM problem, which is set to 5 due to the GPU RAM limit). More details on the models used and training procedures can be found in the supplement. Following convention Käding et al. (2018); Wu (2018); O'Neill et al. (2017), we summarize our DAL performance by the area under curve (AUC) of the error plot. We report the full MSE vs # labeled point plots in the supplement. For the AUC calculation, we use 'sklearn.metrics.auc' Pedregosa et al. (2011) then further normalize by such AUC of random sampling method for easier visualization. All reported results are given in the unit of normalized AUC of MSE ($nAUC_{MSE}$).

## 6 EXPERIMENTAL RESULTS

The performance of all eleven DAL methods on all eight benchmark datasets is summarized in Fig. 3. The y-axis is the normalized $nAUC_{MSE}$, the x-axis is the DAL methods of interest, and the color code represents the different benchmark datasets. The horizontal red dashed line represents the performance of random sampling, which by definition is equal to one (see Section 5). Further details about Fig. 3 are provided in its caption. We next discuss the results, with a focus on findings that are most relevant to DAL in the wild.

The results in Fig. 3 indicate that *all* of our benchmark DAL methods are sensitive to their setting of $\gamma$ - a central hypothesis of this work. As indicated by the vertical bars in Fig. 3, the $nAUC_{MSE}$ obtained by each DAL method varies substantially with respect to $\gamma$. For most of the DAL methods, there exist settings of $\gamma$ (often many) that cause them to perform worse than random sampling. This has significant implications for DAL in the wild since, to our knowledge, there is no general method for estimating a good $\gamma$ setting prior to collecting large quantities of labeled data (e.g., to run trials of DAL with different $\gamma$ settings), and DAL methods may perform worse, and unreliably, when accounting for the uncertainty of $\gamma$.

### 6.1 DALS ARE SENSITIVE TO THEIR POOL RATIO, $\gamma$

The sensitivity of DAL regression models to $\gamma$ may be less significant if there exist $\gamma$ settings that tend to perform well across most problems (for a given DAL method). Fig. 4 presents a histogram of

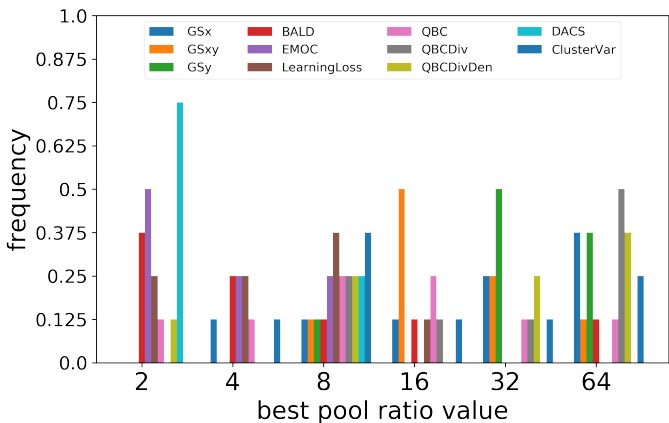

Figure 4: Frequency histogram of the best pool ratio values found in each DAL. For a given DAL method, this figure shows the frequency (% out of 8) that a particular pool ratio (x-axis) performs the best in terms of average $nAUC_{MSE}$ metric.

the best-performing $\gamma$ settings for each DAL method. The results indicate that for each method there is no setting of $\gamma$ that performs best across all problems. This corroborates our observations from the literature where we found a wide range of $\gamma$ settings used across studies. However, we do see that some methods tend to have similar $\gamma$ settings across all problems. For example, DACS has its best performance near $\gamma = 2$, although DACS performs poorly overall. GSxy, however, is one of the best-performing methods overall, and its best-performing settings cluster around $\gamma = 16$. Given this observation, we investigate how well we can perform if we use historical results for a given method to choose a $\gamma$ value for future problems. We emulate this scenario by evaluating the performance of each DAL method when adopting the best single $\gamma$ setting from Fig. 4 (i.e., the setting that wins across the most benchmarks), which we term $\gamma_{prior}$, and then apply it across all benchmarks. The result of this strategy is given by the magenta line in Fig. 3. In most (but not all) cases, $\gamma_{prior}$ yields lower MSE than the average MSE of all $\gamma$ settings (the black lines). In some cases, $\gamma_{prior}$ yields substantial overall performance improvements, such as for GSx and GSxy, suggesting that this is a reasonable $\gamma$ selection strategy, although the benefits seem to vary across DAL models. However, even when using $\gamma_{prior}$, the performance of DAL models still varies greatly, and many models still perform worse than random sampling. Therefore, while $\gamma_{prior}$ may often be beneficial, it does not completely mitigate $\gamma$-uncertainty.

## 6.2 DO ANY DAL METHODS OUTPERFORM RANDOM SAMPLING IN THE WILD?

The results indicate that several DAL methods *tend* to obtain much lower $nAUC_{MSE}$ (i.e., they are better) than random sampling. This includes methods such as GSx, GSxy, GSy, QBC-x (variations of QBC) and ClusterVar. The results therefore suggest that these methods are beneficial more often than not, compared to random sampling - an important property. However, as discussed in Section 6.1, all DAL methods exhibit significant performance variance with respect to $\gamma$, and some of the aforementioned methods still sometimes perform worse than random sampling. For example, this is the case of QBC, GSy, and QBCDivDen on the SINE problem. In settings where DAL is useful, the cost of collecting labels tends to be high, and therefore the risk of poor DAL performance (e.g., relative to simple random sampling) may strongly deter its use. Therefore, another important criteria is performance robustness: do any DAL methods consistently perform better than random sampling, in the wild? Our results indicate that GSx, GSxy, and QBCDiv always perform at least as well as random sampling, and often substantially better, regardless of the problem or $\gamma$ setting. Note that all three robust DALs (GSx, GSxy, QBCDiv) employ x-space diversity in their loss function, which we discuss further in Section 6.3.

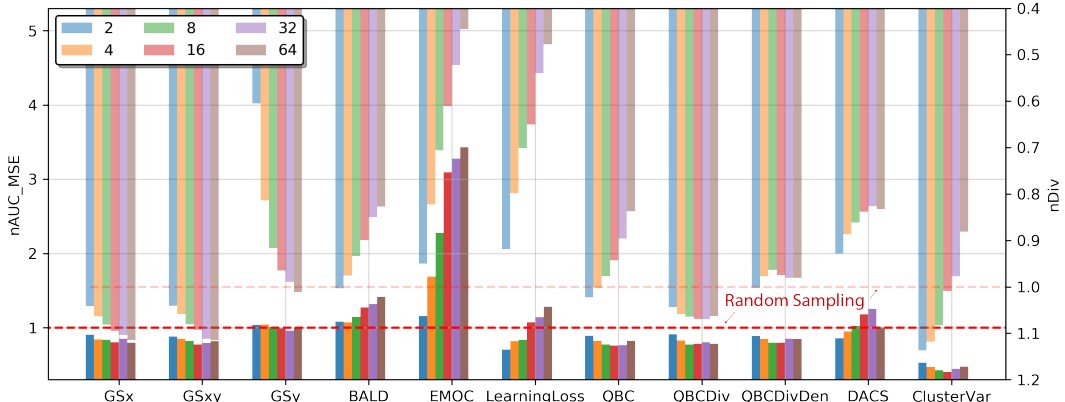

Figure 5: A representative, combined plot with $nAUC_{MSE}$ performance (bottom, y-axis at left, solid) and collapse metric, nDiv (upper, y-axis at right, more transparent) for each of the eleven DAL models at all pool ratios (color coded) for *robotic arm* dataset (ROBO). Dashed horizontal red lines starting from both $y$ axes represent the random sampling's average $nAUC_{MSE}$ and nDiv at 1.

## 6.3 SAMPLE DIVERSITY IS IMPORTANT FOR DAL IN THE WILD

Our results indicate that the best-performing DAL methods are GSx, GSxy, and QBCDiv. We say these methods are "best" because they are both robust (see Section 6.2), and they also usually yield lower MSEs, than other DAL methods. These methods share the common property that they encourage training data diversity, as measured by $x$-space distance between points. Interestingly, GSx *only* relies on x-space diversity. These results suggest that $x$-space diversity is a highly effective DAL acquisition criterion. Furthermore, and in contrast to other criteria, seeking points that maximize $x$-space diversity does not (by definition) increase the risk of mode collapse. Consequently, increasing $\gamma$ results in greater diversity but without any increased risk of mode collapse (more details in Fig. 5). This may be a major reason why GSx, GSxy, and QBCDiv are less sensitive to $\gamma$, and provide much more robust performance in the wild than other DAL methods. While sampling methods that use diversity have been found to be promising Jose et al. (2024), our work provides evidence, for the first time, that sampling based upon diversity *may* be robust to hyperparameter uncertainty (we only examine uncertainty of $\gamma$) while other popular sampling criteria (e.g., estimated model error) seem to be much less reliable in the wild.

To corroborate these findings, we evaluated the $x$-space diversity of each DAL method as a function of $\gamma$. In particular, we calculated the diversity metric as the average nearest neighbor distance

$$Div = \frac{1}{|T|} \sum_{t}^{T} \frac{1}{K} \sum_{i}^{K} \min_{x^* \in \mathcal{Q}^{\mathcal{T}}} dist(x^*, x^i)$$

where $\mathcal{Q}^t$ represents the queried batch at active learning step $t$ and $|T| = 50$ is the total number of active learning steps. Note that this metric is similar to, but not a simple average of $q_{GSx}(x)$ as $Div$ only focuses on per batch diversity and does not take the labeled set into consideration. It is also further normalized ($nDiv$) by the value of random sampling for each dataset separately. The lower this metric's value, the more severe the mode collapse issue would be.

The $nDiv$ is plotted in the top half of Fig. 5 using the inverted right y-axis. For the obvious failure cases (BALD, EMOC and Learning Loss) in this particular dataset (their $nAUC_{MSE}$ exceeds 1), a clear trend of mode collapse can be observed in the upper half of the plot (nDiv much lower than 1). Meanwhile, a strong correlation between the pool ratio and the diversity metric can be observed: (i) For GSx and GSxy methods, which seek to maximize diversity, their diversity increases monotonically with larger pool ratio. (ii) For uncertainty-based methods (BALD, EMOC, LearningLoss, QBC, MSE), which seek to maximize query uncertainty, their diversity decreases monotonically with larger pool ratios. (iii) For combined methods like QBCDiv and QBCDivDen, the relationship between pool ratio and diversity shows a weak correlation, consistent with the benefits of having diversity

as a selection criterion. (iv) Lastly, we observe that all top-performing methods have high diversity, regardless of $\gamma$, suggesting it is an important condition for effective DAL.

## 7 CONCLUSIONS

For the first time, we evaluated eleven state-of-the-art DAL methods on eight benchmark datasets for regression *in the wild*, where we assume that the best pool ratio hyperparameter, $\gamma$, is uncertain. We summarize our findings as follows:

- *DAL methods for regression often perform worse than simple random sampling, when evaluated in the wild*. Using $\gamma$ as an example, we systematically demonstrate the rarely-discussed problem that most DAL models are often outperformed by simple random sampling when we account for HP uncertainty.
- *Some DAL methods were relatively robust, and outperformed random sampling robustly in the wild (e.g., GSx, GSxy, QBCDiv).*
- *Insofar as robustness to pool ratio is concerned, our results suggest that DAL approaches utilizing sample diversity tend to be much more robust in the wild than other popular selection criteria.*

### 7.1 LIMITATIONS

One limitation of this work is that we focused on scientific computing benchmark problems, and problems with relatively low dimensionality. Including higher dimensional problems is an especially important opportunity for future work due to the importance of vision problems in the DAL community, and also because sensitivity to pool ratio has been noted in that setting as well Yoo & Kweon (2019); Sener & Savarese (2017), but not studied systematically. Another important limitation is that we constrained our evaluation of DAL methods to uncertainty in their pool ratio. Future studies would benefit from evaluating each DAL approach with respect to uncertainty in all of its relevant DAL HPs (i.e., those that require labeled data to be optimized), providing a more comprehensive assessment of modern DAL methods in the wild.

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
