# SUPPLEMENT: DOES DEEP ACTIVE LEARNING WORK IN THE WILD?

## 1 DETAILS OF THE BENCHMARKING METHODS

**Core-set (GSx: Greedy sampling in $x$ space)** Sener & Savarese (2017). This approach only relies upon the diversity of points in the input space, $\mathcal{X}$, when selecting new query locations. A greedy selection criterion is used, given by

$$q_{GSx}(x^*) = \min_{x \in \mathcal{L} \cup \mathcal{Q}} dist(x^*, x)$$

where $\mathcal{L}$ is the labeled set, $\mathcal{Q}$ is the already selected query points and $dist$ being L2 distance.

**Greedy sampling in $y$ space (GSy)** Wu et al. (2019). Similar to GSx which maximizes diversity in the $x$ space in a greedy fashion, GSy maximizes the diversity in the $y$ space in a greedy fashion:

$$q_{GSy}(x^*) = \min_{y \in \mathcal{L} \cup \mathcal{Q}} dist(f(x^*), y)$$

where $f(x)$ is the current model prediction of the $x$ and $y$ is the labels in the already labeled training set plus the predicted labels for the points (to be labeled) selected in the current step.

**Greedy sampling in xy space (GSxy)** Wu et al. (2019). Named as 'Improved greedy sampling (iGS)' in the original paper Wu et al. (2019), this approach combines GSx and GSy and uses multiplication of the distance of both $x$ and $y$ space in its acquisition function:

$$q_{GSxy}(x^*) = \min_{(x,y) \in \mathcal{L} \cup \mathcal{Q}} dist(x^*, x) * dist(f(x^*), y)$$

**Query-by-committee (QBC)** Seung et al. (1992) The QBC approach is pure uncertainty sampling if we set $q(x) = q_{QBC}(x)$:

$$q_{QBC}(x) = \frac{1}{N} \sum_{n=1}^{N} (\hat{f}_n(x) - \mu(x))^2$$

Here $\hat{f}_n$ denotes the $n^{th}$ model in an ensemble of $N_{ens}$ models (DNNs in our case), and $\mu(x)$ is the mean of the ensemble predictions at $x$. In each iteration of AL these models are trained on all available training data at that iteration.

**QBC with diversity (Div-QBC)** Kee et al. (2018). This method improves upon QBC by adding a term to $q$ that also encourages the selected query points to be diverse from one another. This method introduces a hyperparameter for the relative weight of the diversity and QBC criteria and we use an equal weighting ($\alpha = 0.5$ Kee et al. (2018)).

$$q_{QBCDiv}(x) = (1 - \alpha) * q_{QBC}(x) + \alpha * q_{div}(x)$$
$$q_{div}(x^*) = q_{GSx}(x^*)$$

**QBC with diversity and density (DenDiv-QBC)** Kee et al. (2018). This method builds upon Div-QBC by adding a term to $q(x)$ that encourages query points to have uniform density. This method introduces two new hyperparameters for the relative weight ($\alpha = \beta = \frac{1}{3}$) of the density, diversity, and QBC criteria, and we use an equal weighting as done in the original paper Kee et al. (2018).

$$q_{QBCDivDen}(x) = (1 - \alpha - \beta) * q_{QBC}(x)$$
$$+ \alpha * q_{div}(x) + \beta * q_{den}(x)$$
$$q_{den}(x^*) = \frac{1}{k} \sum_{x \in N_k(x^*)} sim(x^*, x)$$

where $N_k(x^*)$ is the k nearest neighbors of an unlabeled point, $sim(x^*, x)$ is the cosine similarity between points.

**Bayesian active learning by disagreement (BALD)** Tsymbalov et al. (2018). BALD uses the Monte Carlo dropout technique to produce multiple probabilistic model output to estimate the uncertainty of model output and uses that as the criteria of selection (same as $q_{QBC}(x)$). We used 25 forward passes to estimate the disagreement.

**Expected model output change (EMOC)** Käding et al. (2018); Ranganathan et al. (2020). EMOC is a well-studied AL method for the classification task that strives to maximize the change in the model (output) by labeling points that have the largest gradient. However, as the true label is unknown, some label distribution assumptions must be made. Simple approximations like uniform probability across all labels exist can made for classification but not for regression tasks. Ranganathan et al. (2020) made an assumption that the label is simply the average of all predicted output in the unlabeled set ($y'(x') = \mathbb{E}_{x \in \mathcal{U}} f(x)$) and we use this implementation for our benchmark of EMOC.

$$q_{EMOC}(x') = \mathbb{E}_{y'|x'} \mathbb{E}_x ||f(x; \phi') - f(x; \phi)||_1$$
$$\approx \mathbb{E}_x ||\nabla_\phi f(x; \phi) * \nabla_\phi \mathcal{L}(\phi; (x', y'))||_1$$

where $f(x; \phi)$ is the current model output for point $x$ with model parameter $\phi$, $\phi'$ is the updated parameter after training on labeled point x' with label y' and $\mathcal{L}(\phi; (x', y')$ is the loss of the model with current model parameter $\phi$ on new labeled data $(x', y')$.

**Learning Loss** Yoo & Kweon (2019). Learning Loss is another uncertainty-based AL method that instead of using proxies calculated (like variance), learns the uncertainty directly by adding an auxiliary model to predict the loss of the current point that the regression model would make. The training of the auxiliary model concurs with the main regressor training and it uses a soft, pair-wise ranking loss instead of Mean Squared Error (MSE) loss to account for the fluctuations of the actual loss during the training.

$$q_{LL}(x) = f_{loss}(x)$$

where $f_{loss}(x)$ is the output of the loss prediction auxiliary model. In this AL method, there are multiple hyper-parameters (co-training epoch, total auxiliary model size, auxiliary model connections, etc.) added to the AL process, all of which we used the same values in the paper if specified Yoo & Kweon (2019).

**Density Aware Core-set (DACS)** Kim & Shin (2022) A diversity-based AL method that not only considers core-set metric but also considers the density and strives to sample low-density regions. The original DACS also encodes the image space into feature space and uses locality-sensitive hashing techniques to accelerate the nearest neighbor calculation and prevent computational bottlenecks. As our scientific computing tasks does not involve high dimensional image as well as having much lower dataset size in general, instead of encoded feature space distance, we used input space distance and locality-sensitive hashing was dropped as we don't face such computational bottleneck for the nearest neighbor calculation with our smaller pool compared to theirs.

**Cluster Margin adapted to regression problem: Cluster Variance (ClusterVar)** Citovsky et al. (2021) To alleviate the robustness issue arising in larger batch AL scenarios, Cluster-Margin Citovsky et al. (2021) method is proposed to add necessary diversity to the uncertainty sampling. The original method used margin as its uncertainty metric as it was demonstrated on image classification tasks, and we adapted it into a variance metric in a regression setting. During Cluster Margin, Hierarchical Agglomerative Clustering (HAC) is run once on the unlabeled pool before the AL process and during each round a round-robin selection is carried out from the smallest cluster to the largest cluster, each time selecting the unlabeled sample with the highest uncertainty metric.

## 2 DETAILS OF BENCHMARK DATASETS USED

**1D sine wave (Wave).** A noiseless 1-dimensional sinusoid with varying frequency over $x$, illustrated in Fig. 1.

$$y = x * sin(a_1 * sin(a_2 * x)),$$

where $a_1 = 3$ and $a_2 = 30$ is chosen to make a relative complicated loss surface for the neural network to learn while also having a difference in sensitivity in the domain of x.

**2D robotic arm (Arm) Ren et al. (2020)** In this problem we aim to predict the 2D spatial location of the endpoint of a robotic arm based on its joint angles. Illustrated in Fig. 1. The Oracle function is given by

$$y_0 = \sum_{i=1}^{3} \cos(\frac{pi}{2}x_i) * l_i, y_1 = x_0 + \sum_{i=1}^{3} \sin(\frac{pi}{2}x_i) * l_i$$

where $y$ is the position in the 2D plane, $x_0$ is the adjustable starting horizontal position, $x_{i=1,2,3}$ are the angles of the arm relative to horizontal reference and $l_{i=0,1,2} = [0.5, 0.5, 1]$ represents the i-th length of the robotic arm component. The dataset is available under the MIT license.

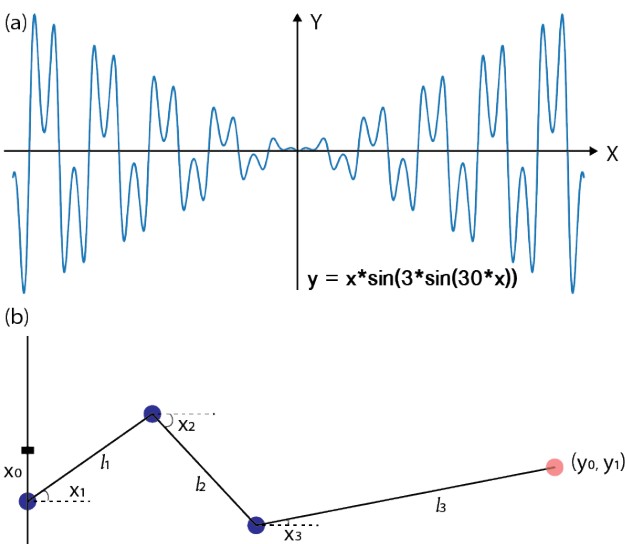

Figure 1: Schematic illustration of sine wave and robotic arm datasets

**Stacked material (Stack) Chen et al. (2019).** In this problem, we aim to predict the reflection spectrum of a material, sampled at 201 wavelength points, based upon the thickness of each of the 5 layers of the material, illustrated in Fig. 2. It was also benchmarked in Ren et al. (2022). An analytic Oracle function is available based upon physicsChen et al. (2019).

**Artificial Dielectric Material (ADM) Deng et al. (2021b)** This problem takes the geometric structure of a material as input, and the reflection spectrum of the material, as a function of frequency, illustrated in Fig. 2. It was also benchmarked in Deng et al. (2021a). This dataset -released under CC BY 4.0 License - consists of input space of 3D geometric shape parameterized into 14 dimension space and the output is the spectral response of the material. The oracle function is a DNN Deng et al. (2021a).

**NASA Airfoil (Foil) Dua & Graff (2017)** NASA dataset published on `https://archive.ics.uci.edu/dataset/291/airfoil+self+noise` UCI ML repository under CC BY 4.0 License Dua & Graff (2017) obtained from a series of aerodynamic and acoustic tests of 2D/3D airfoil blade sections conducted in an anechoic wind tunnel, illustrated in Fig. 3. The input is the physical properties of the airfoil, like the angle of attack and chord length and the regression target is the sound pressure in decibels. We use a well-fitted random forest fit to the original dataset as our oracle function following prior workTrabucco et al. (2022). The fitted random forest architecture and its weights are also shared in our code repo to ensure future work makes full use of such benchmark datasets as we did.

**Hydrodynamics (Hydro) Dua & Graff (2017)** Experiment conducted by the Technical University of Delft, illustrated in Fig. 3, (hosted on `https://archive.ics.uci.edu/ml/datasets/Yacht+Hydrodynamics` UCI ML repository under CC BY 4.0 License Dua & Graff (2017)), this dataset contains basic hull dimensions and boat velocity and their corresponding residuary resistance. Input is 6 dimensions and output is the 1 dimension. We use a well-fitted random forest fit to the original dataset as our oracle function. The fitted random forest architecture and its weights are also

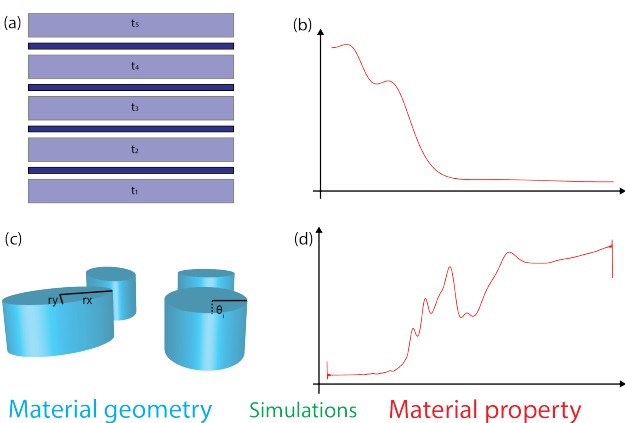

Figure 2: (a, c) are schematic illustration of two material design datasets (Stack & ADM). (b, d) are example spectra of their material property after simulations from their geometric parameterization (typically from Maxwell equation solvers that are slow and hence can benefit from active learning)

shared in our code repository to ensure future work makes full use of such benchmark dataset as we did.

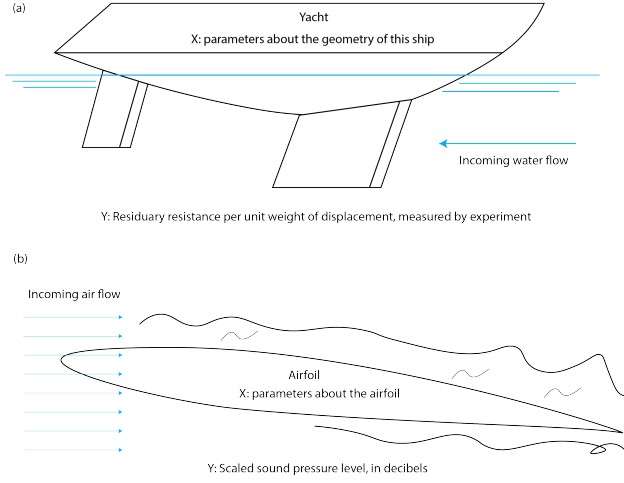

Figure 3: Schematic illustration of Airfoil and Hydro experiments. Reproduced from the original source of experiment reports from NASA and Delft University of technology. (a) The Hydro experiment with an actual yacht being built and resistance was measured in a water flow experiment as the regression target y. (b) Airfoil experiment where input is the parameters of the airfoil and the sound pressure level is measured as target $y$ of the regression task.

**Bessel equation** The solution to the below single dimension second-order differential equation:

$$x^2 \frac{d^2y}{dx^2} + x \frac{dy}{dx} + (x^2 - \alpha^2)y = 0$$

where input is $\alpha$ and $x$ position given. $\alpha$ is limited to non-negative integers smaller than 10 and $x \in [0, 10]$. The solution examples can be visualized in Fig. 4. Our choice of $\alpha$ values makes the Bessel functions cylinder harmonics and they frequently appear in solutions to Laplace's equation (in cylindrical systems). The implementation we used is the python package 'scipy.special.jv(v,z)' Virtanen et al. (2020).

**Damping oscillator equation** The solution to the below ordinary second-order differential equation:

$$m\frac{dx^2}{d^2t} + b\frac{dx}{dt} + \frac{mg}{l}x = 0$$

where m is the mass of the oscillator, b is the air drag, g is the gravity coefficient, l is the length of the oscillator's string and it has analytical solution of form

$$x = ae^{-bt}cos(\alpha - \psi)$$

where a is the amplitude, b is the damping coefficient, $\alpha$ is the frequency and $\psi$ is the phase shift. We assume $\psi$ to be 0 and let a,b,$\alpha$ be the input parameters. The output, unlike our previous ODE dataset, is taken as the first 100 time step trajectory of the oscillator, making it a high dimensional manifold (nominal dimension of 100 with true dimension of 3). The trajectory is illustrated in Fig. 4. We implement the above solution by basic python math operations.

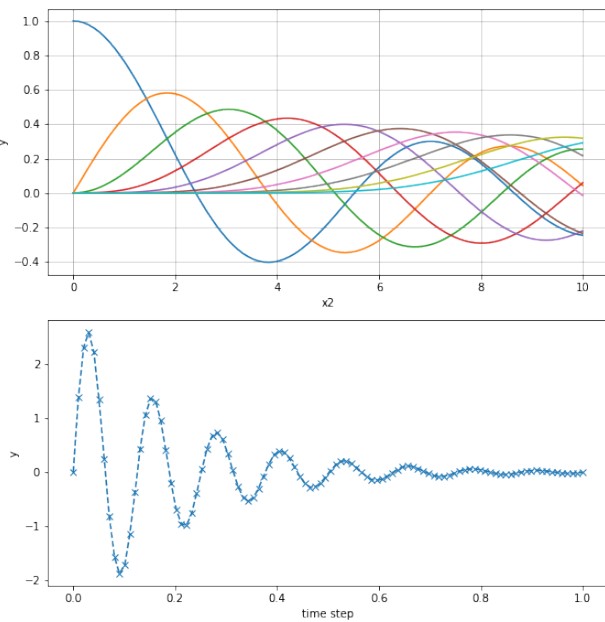

Figure 4: Schematic illustration of Bessel function solution and the damping oscillator solutions.

## 3 LIST OF POOL RATIO USED IN EXISTING LITERATURE

17000 McCallumzy & Nigamy (1998), 20 to 2000 Kee et al. (2018), 300 to 375Santos et al. (2020), 11-20 Roy et al. (2018), 1000 Burbidge et al. (2007), and 1 to 11 Tan et al. (2019).

## 4 DETAILS OF MODELS TRAINING AND ARCHITECTURE

In the below Table 1, we present the model architecture for each of our benchmarked datasets. Unless otherwise noted, all of them are fully connected neural networks.

Table 1: Regression model, $\hat{f}$ architecture details for each problem. *: for ADM, there are 3 layers of convolutions after the linear layer)

| FEAT | SINE | ROBO | STACK | ADM | FOIL | HYDR | BESS | DAMP |
|------|------|------|-------|-----|------|------|------|------|
| NODE | 20 | 500 | 700 | 1500 | 200 | 50 | 50 | 500 |
| LAYER | 9 | 4 | 9 | 4* | 4 | 6 | 6 | 6 |

We implemented our models in PyTorch Paszke et al. (2019). Beyond the above architectural differences, the rest of the model training settings are the same across the models: Starting labeled set of size 80, in each step DAL finds 40 points to be labeled for 50 active learning steps. Each regression model is an ensemble network of 10 models of size illustrated in Table 1 except the ADM dataset (5 instead of 10 due to RAM issue). The test dataset is kept at 4000 points uniformly sampled across the $x$-space and they are fixed the same across all experiments for the same dataset. No bootstrapping is used to train the ensemble network and the only source of difference between networks in the ensemble (committee) is the random initialization of weights.

The batch size is set to be 5000 (larger than the largest training set) so that the incomplete last batch would not affect the training result (as we sample more and more data, we can't throw away the last incomplete batch but having largely incomplete batch de-stabilizes training and introduce noise into the experiment. Adam optimizer is used with 500 training epochs and the model always retrains from scratch. (We observe that the training loss is much higher if we inherit from the last training episode and do not start from scratch, which is consistent with other literature Beck et al. (2021)). The learning rate is generally 1e-3 (some datasets might differ), and the decay rate of 0.8 with the decay at the plateau training schedule. The regularization weight is usually 1e-4 (dataset-dependent as well). The hyper-parameters only change with respect to the dataset but never with respect to DAL used.

The hyperparameters are tuned in the same way as the model architecture: Assume we have a relatively large dataset (2000 randomly sampling points) and tune our hyperparameter on this set. This raises another robustness problem of deep active learning, which is how to determine the model architecture before we have enough labels. This is currently out of the scope of this work as we focused on how different DALs behave with the assumption that the model architectures are chosen smartly and would be happy to investigate this issue in future work.

For the BALD method, we used a dropout rate of 0.5 as advised by previous work. As BALD requires a different architecture than other base methods (a dropout structure, that is capable of getting a good estimate even with 50% of the neurons being dropped), the model architecture for the active learning is different in that it enlarges each layer by a constant factor that can make it the relatively same amount of total neurons like other DAL methods. Initially, the final trained version of the dropout model is used as the regression model to be evaluated. However, we found that an oversized dropout model hardly fits as well as our ensembled counterpart like other DAL methods. Therefore, to ensure the fairness of comparison, we trained another separate, ensembled regression model same as the other DALs and reported our performance on that.

For the LearningLoss method, we used the same hyper-parameter that we found in the cited work in the main text: relative weight of learning loss of 0.001 and a training schedule of 60% of joint model training and the rest epoch we cut the gradient flow to the main model from the auxiliary model. For the design of the auxiliary model, we employed a concatenation of the output of the last three hidden layers of our network, each followed by a fully connected network of 10 neurons, before being directed to the final fully connected layer of the auxiliary network that produces a single loss estimate.

For the EMOC method, due to RAM limit and time constraint, we can not consider all the model parameters during the gradient calculation step (For time constraint, Table 2 gives a good reference of how much longer EMOC cost, even in this reduced form). Therefore, we implemented two approximations: (i) For the training set points where the current model gradients are evaluated, instead of taking the ever-growing set that is more and more biased towards the DAL selection, we fixed it to be the 80 original, uniformly sampled points. (ii) We limit the number of model parameters to evaluate the EMOC criteria to 50k. We believe taking the effect of 50 thousand parameters gives a good representation of the model's response (output change) for new points. We acknowledge that these approximations might be posing constraints to EMOC, however, these are practical, solid challenges for DAL practitioners as well and these are likely the compromise to be made during application.

## 4.1 COMPUTATIONAL RESOURCES

Here we report the computational resources we used for this work: AMD CPU with 64 cores; NVIDIA 3090 GPU x4 (for each of the experiment we used a single GPU to train); 256GB RAM.

# 5 ADDITIONAL PERFORMANCE PLOTS

As the benchmark conducts a huge set of experiments that are hard to fit in the main text, here we present all the resulting figures for those who are interested to dig more takeaways.

## 5.1 TIME PERFORMANCE OF THE BENCHMARKED DAL METHODS

We also list the time performance of each DAL method, using the ROBO dataset as an example in the below Table 2. Note that this is only the sampling time cost, not including the model training time, which is usually significantly larger than the active learning sampling time at each step. The only DAL method that potentially has a time performance issue is the EMOC method, which requires the calculation of each of the gradients with respect to all parameters and therefore takes a much longer time than other DAL methods. However, as it is shown in the main text that it is not a robust method in our setting, there is no dilemma of performance/time tradeoff presented here.

Table 2: Time performance for average time spent during the sampling process for ROBO dataset per active learning step (40 points) in ms for pool ratio of 2. LL: LearningLoss

| DATASET | RANDOM | GSX | GSXY | GSY | BALD | EMOC |
|---------|--------|-----|------|-----|------|------|
| TIME    | 2.15   | 4.96 | 10.27 | 6.85 | 9.06 | **756.6** |
| DATASET | LL     | QBC | QBCDIV | QBCDIVDEN | DACS | CLUSTERVAR |
| TIME    | 6.53   | 4.29 | 9.04 | 10.38 | 21.70 | 4.31 |

## 5.2 COMBINED PLOT WITH $nAUC_{MSE}$ AND NDIV

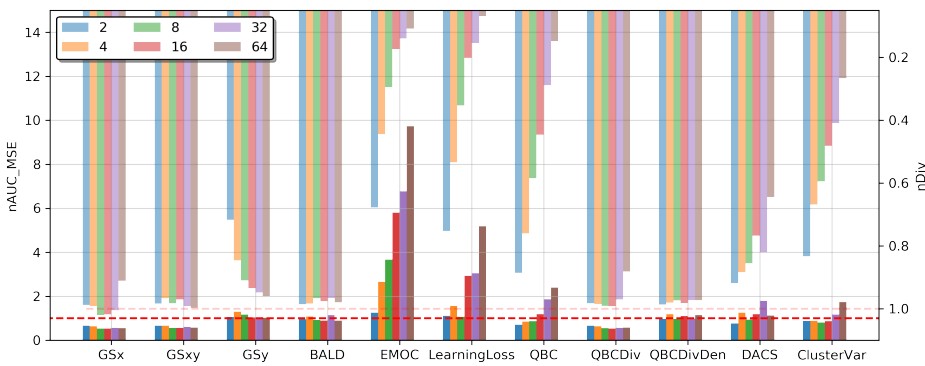

Figure 5: $nAUC_{MSE}$ and nDiv plot for SINE

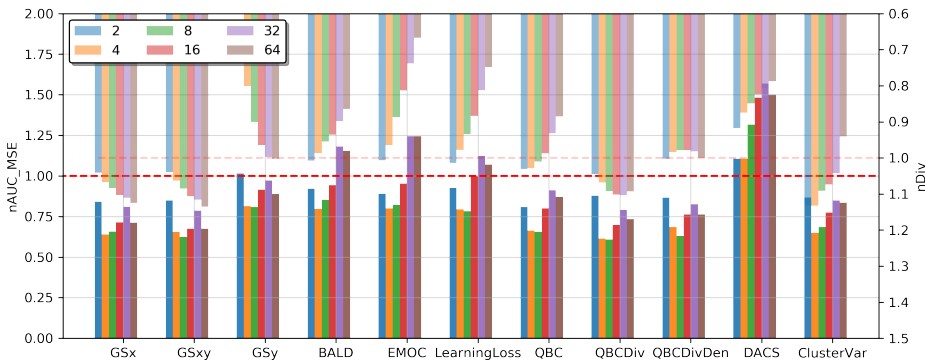

Figure 6: $nAUC_{MSE}$ and nDiv plot for STACK

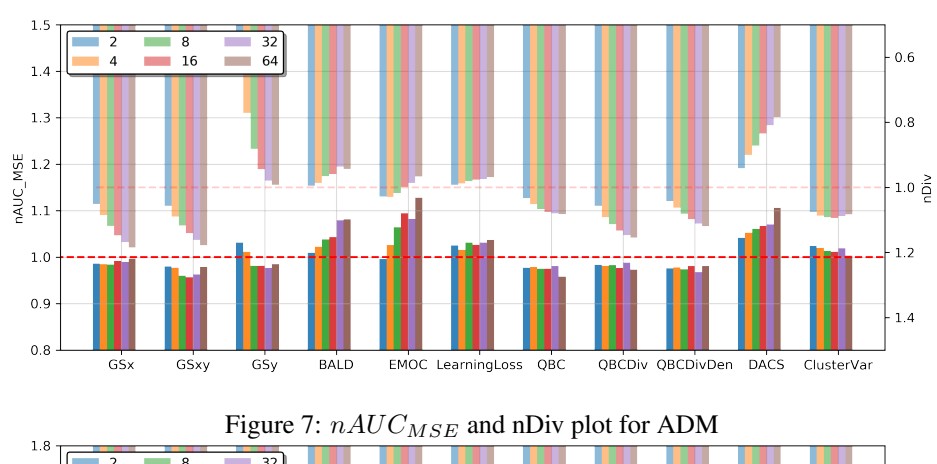

Figure 7: $nAUC_{MSE}$ and nDiv plot for ADM

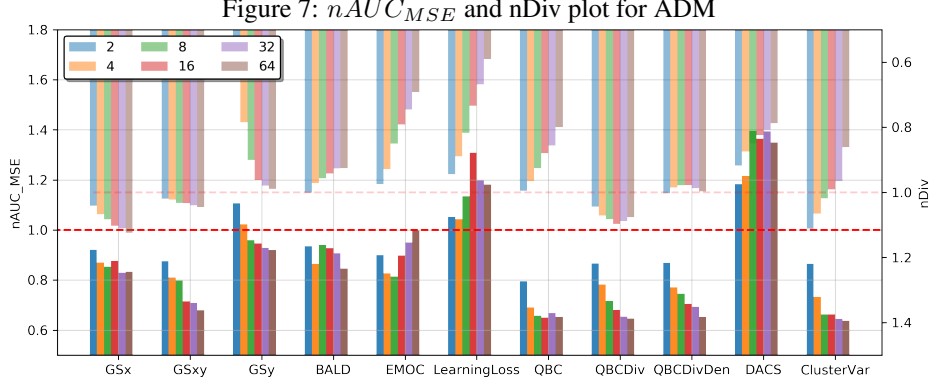

Figure 8: $nAUC_{MSE}$ and nDiv plot for FOIL

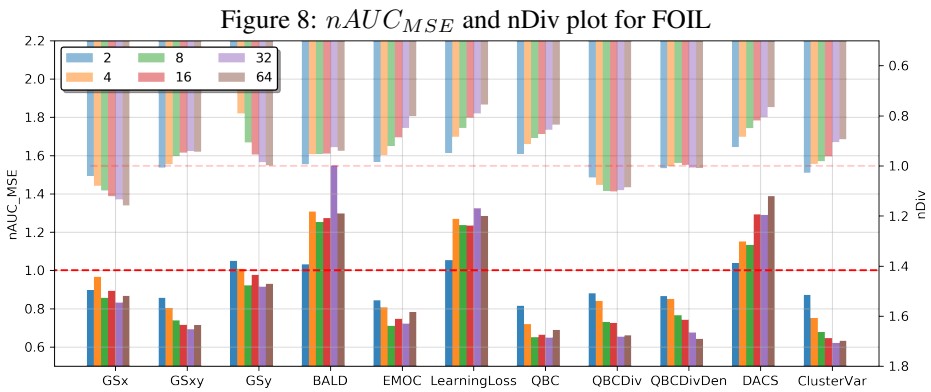

Figure 9: $nAUC_{MSE}$ and nDiv plot for HYDR

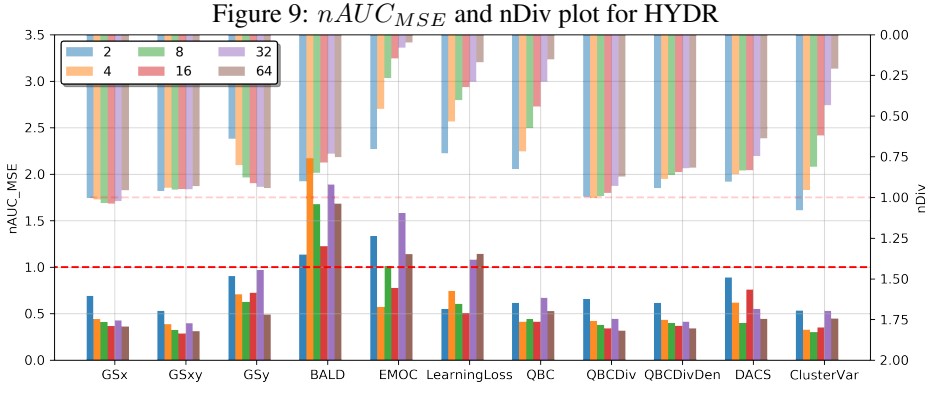

Figure 10: $nAUC_{MSE}$ and nDiv plot for BESS

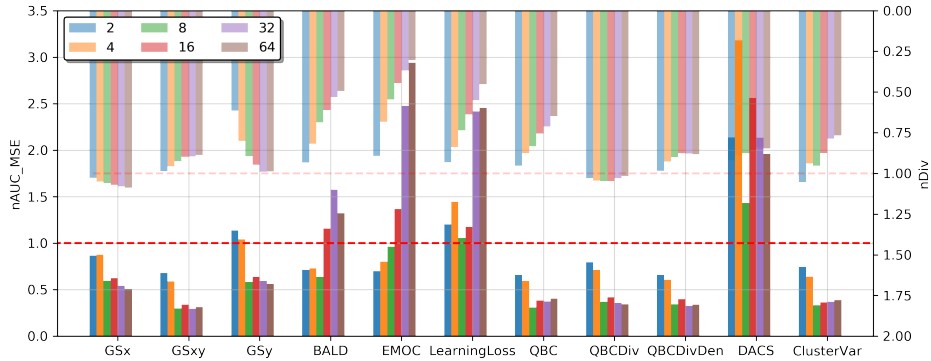

Figure 11: $nAUC_{MSE}$ and nDiv plot for DAMP

## 5.3 MSE VS ACTIVE LEARNING STEP PLOT

We also present the traditional plot of the MSE vs active learning step for reference. For each of the plots below, the MSE are smoothed with a smoothing parameter of 0.5 using the tensorboard smoothing visualizing function Abadi et al. (2015). The $x$ labels are from 0 - 49, where 0 measures the end of the first active learning step.

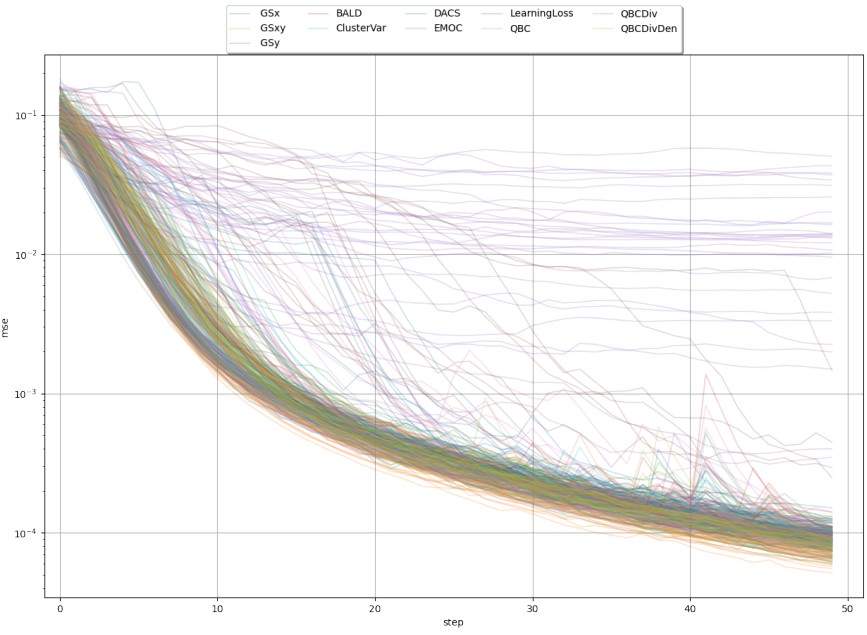

Figure 12: MSE plot for SINE

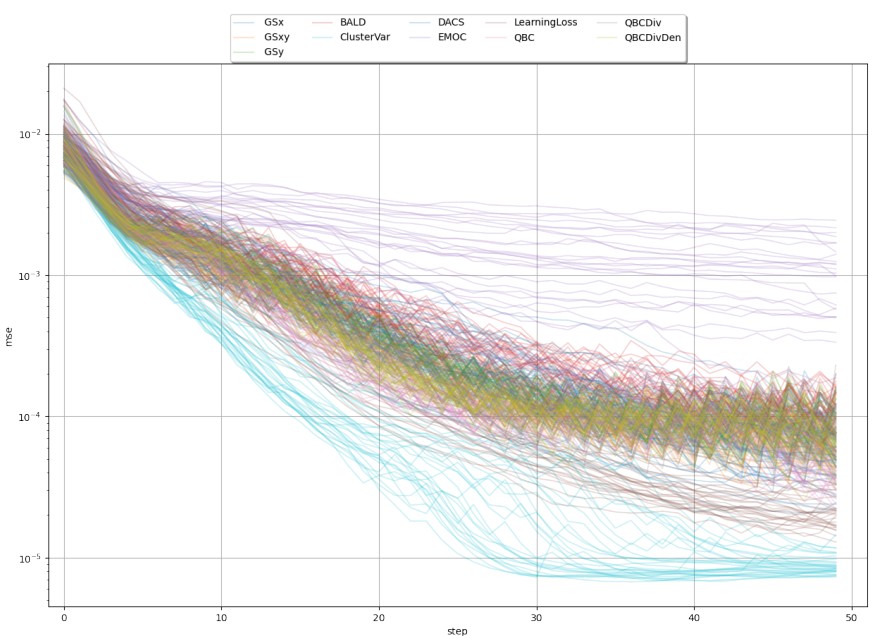

Figure 13: MSE plot for ROBO

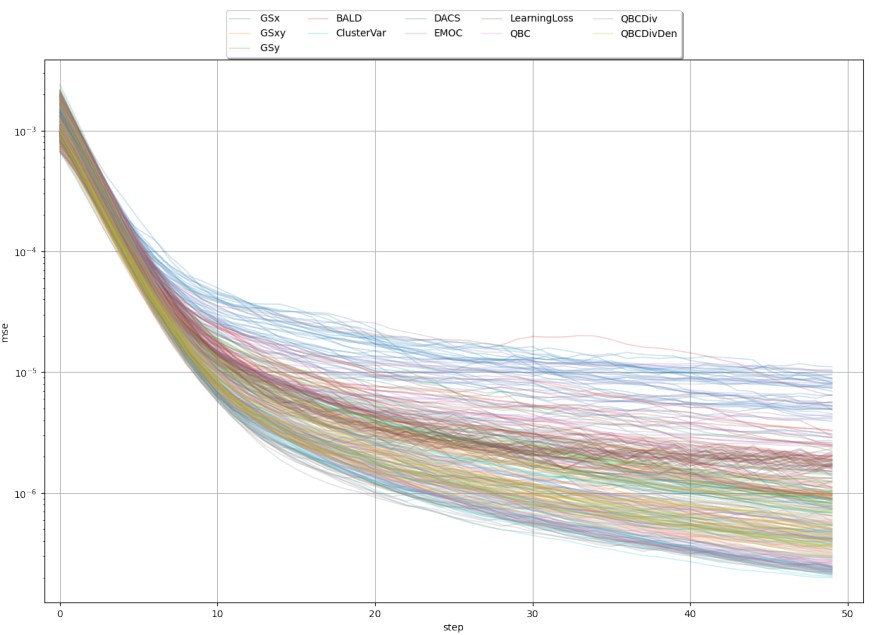

Figure 14: MSE plot for STACK

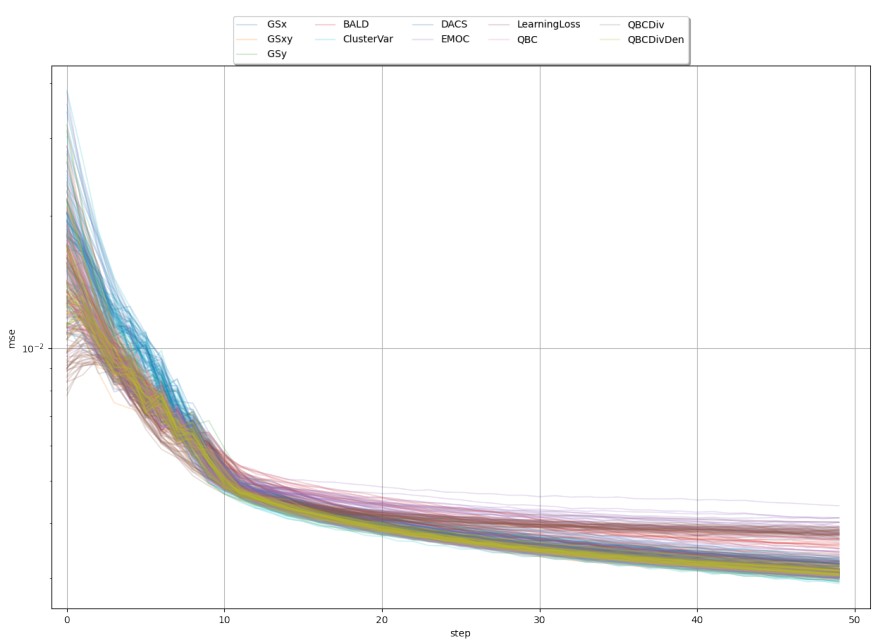

Figure 15: MSE plot for ADM

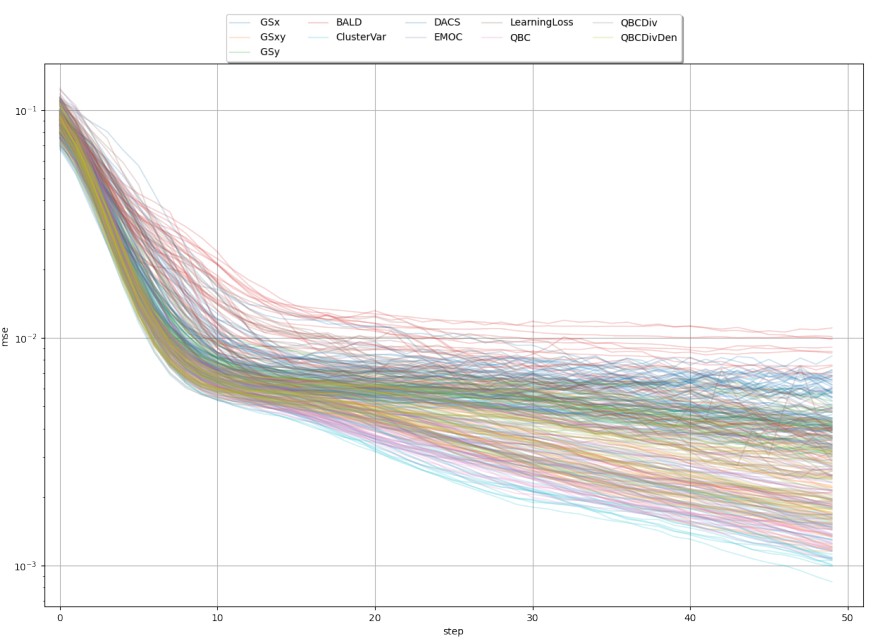

Figure 16: MSE plot for HYDR

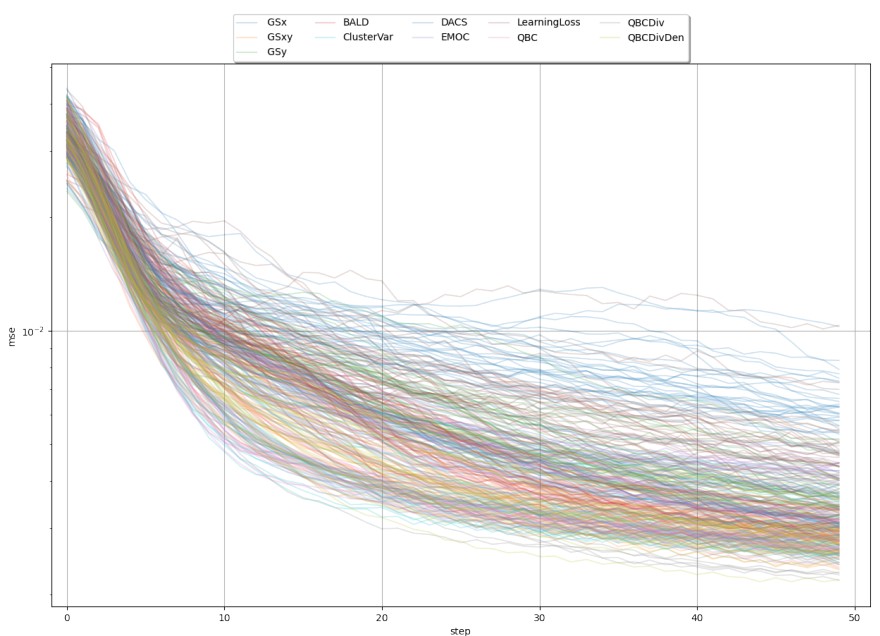

Figure 17: MSE plot for FOIL

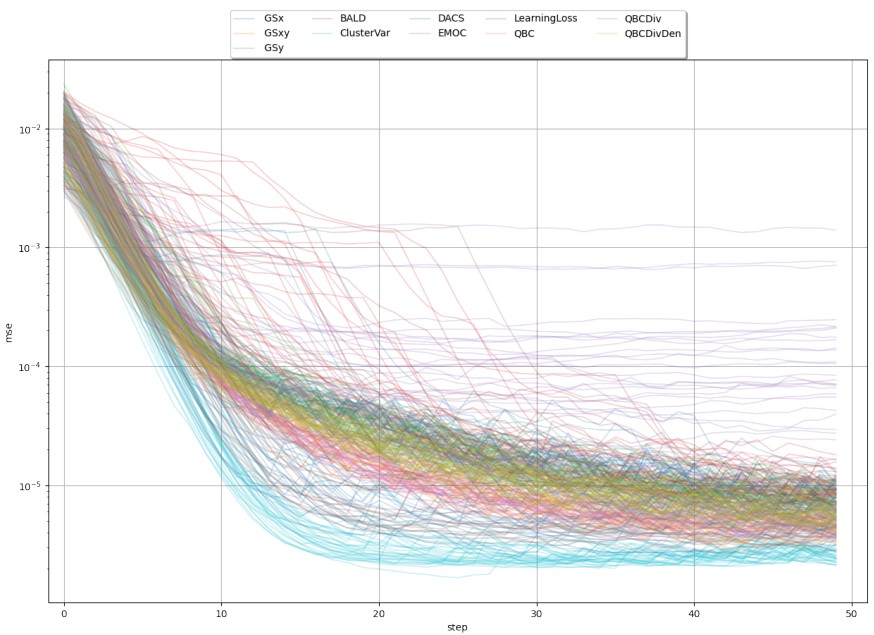

Figure 18: MSE plot for BESS

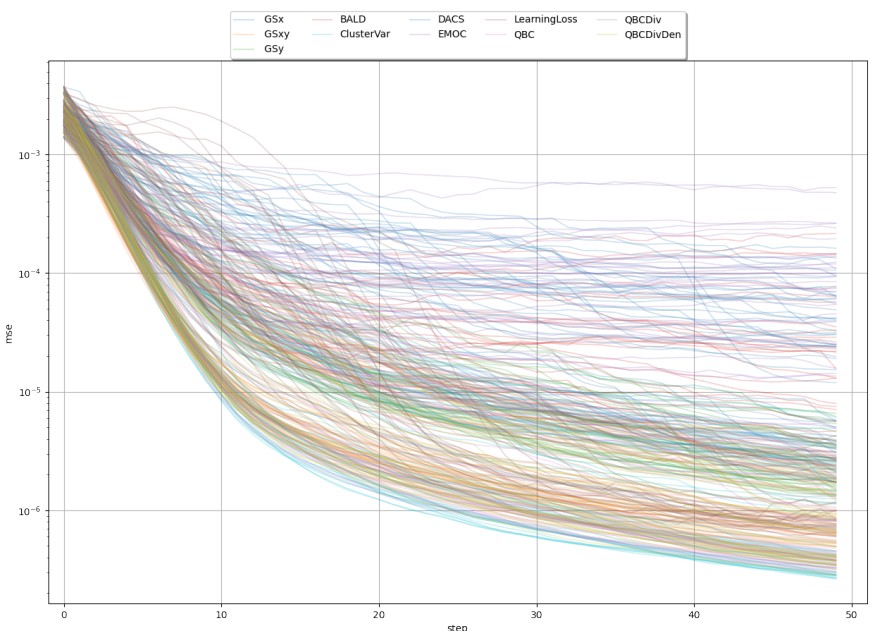

Figure 19: MSE plot for DAMP