# OpenReview forum: "Does Deep Active Learning Work in the Wild?"
_ICLR.cc/2025/Conference — ICLR 2025 Conference Withdrawn Submission_

### Official Review · Reviewer_GVoN · 2024-10-28

**Soundness:** 2
**Presentation:** 3
**Contribution:** 2
**Rating:** 3
**Confidence:** 3

**Summary:**

This paper investigates the robustness of deep active learning (DAL) methods in real-world applications, where optimal hyperparameter (HP) settings are not predetermined and may be challenging to optimize without compromising labeling efficiency. Typically, studies assume known or optimized HP settings, yet real-world scenarios often involve significant HP uncertainty. The authors assess eleven modern DAL methods across eight benchmark datasets, focusing on a single, impactful HP: the pool ratio.

**Strengths:**

1. This thesis examines one of the interesting and equally important questions-“Does Deep Active Learning Work in the Wild?”.
2. The authors have done extensive experiments to demonstrate the importance of the hyperparameter pool ratio.

**Weaknesses:**

1. The question of whether Deep Active Learning works in real-world settings is indeed important, and this study addresses it primarily by examining the effect of hyperparameters. While this focus on hyperparameters is valuable, it is not fundamentally different from other machine learning tasks, which are also often sensitive to hyperparameter settings. Consequently, the title of the work appears somewhat ambitious, as the actual scope of the study is relatively narrow and could be seen as slightly overstated.

2. Building on the previous critique, even if the focus is limited to hyperparameters as the primary issue, it would be more impactful if the authors proposed solutions or developed a hyperparameter-sensitive DAL method. Simply highlighting the problem and repeatedly evaluating existing methods falls short of advancing the field and may not meet the contribution standard expected at venues like ICLR.

3. The fact that DAL in the wild performs even less well than random samples is a good motivation, and should not just be a conclusion. In my opinion, it should be more the beginning of the paper than the end.

**Questions:**

I have no questions regarding the authors' assessment of hyperparameter sensitivity. However, following this evaluation, I would like the authors to propose a solution to address hyperparameter sensitivity rather than simply presenting the issue.

---

### Official Review · Reviewer_G6AZ · 2024-11-01

**Soundness:** 2
**Presentation:** 2
**Contribution:** 2
**Rating:** 3
**Confidence:** 3

**Summary:**

In this paper, the authors evaluate the performance of eleven deep active learning methods on eight benchmark problems ('in the wild' settings) , by varying a key hyper-parameter represented by the the pool ratio, i.e. the ratio between the total number of unlabeled samples and the budget (number of samples to be labeled in each cycle). Despite adjusting only one hyper-parameter, the results indicate that eight of the eleven DAL methods sometimes under-perform relative to simple random sampling and some frequently perform worse. Only three methods always outperform random sampling.

**Strengths:**

- The paper is relatively clearly written and easy to follow
- Active Learning represents an interesting research area, taking into account that nowadays the amount of data being generated is so huge, it is impossible to be human-labeled.

**Weaknesses:**

- There is no scientific contribution
- The evaluation is limited. The paper analysis only one hyper-parameter. A more comprehensive analysis would have been preferred and expected
- The evaluated methods are old (only two of them are 4 and respectively 5 years old), before 2020.

**Questions:**

Here are my concerns:
- The hyper-parameter (gamma - the pool ratio) the authors use in their study is defined too late (section 3.3). It should be defined in the Introduction.
- Their definition of the hyper-parameter is mis-leading. Actually, the parameter the authors refer to is the annotation budget, i.e the number of samples extracted from the unlabeled data pool to be labeled in each cycle. The authors are asked to modify this aspect in the paper.
- There is another recent related work to the current one, namely (Holzmüller et al., 2023). The authors should very clearly state, providing more details, what is the difference between their work and the aforementioned reference
- The references in section 2.2 should be updated (the most recent one is from 2021)
- The authors should enrich their study, by taking into account another relevant hyper-parameter for active learning, the initiai ratio between labeled and unlabeled samples.
- Last, the authors should include in their study more recent approaches.

---

### Official Review · Reviewer_3Z6P · 2024-11-01

**Soundness:** 1
**Presentation:** 2
**Contribution:** 1
**Rating:** 3
**Confidence:** 3

**Summary:**

This article raises the issue of uncertainty concerning the choices of hyperparameters when deploying DAL in real-world applications. The content is centered around the pool ratio, a hyperparameter that determines the amount of unlabeled data investigated when querying new samples to be labeled. Its influence is demonstrated in a study comprising 8 regression problems and 10 DAL strategies. The main line of reasoning is based on the fact that in real applications, hyperparameters such as the pool ratio cannot be optimized, and the varying performances of different strategies exhibit issues related to their application in real problems. Furthermore, the work distills three DAL strategies with robust performances in the choice of pool ratio, promoting their application. In addition, the work points out that the key ingredient to successful DAL strategies "in the wild" is a diversity component.

**Strengths:**

The article focuses on the critical transition between theoretical research and practical application of DAL, a problem of great importance in DAL research. In addition to empirical evidence about which strategies perform best, the authors investigate critical aspects that make these strategies superior.

**Weaknesses:**

There is a significant discrepancy between the motivation of the topic and the subsequent investigation. While choosing DAL hyperparameters in the application is critical, the authors investigate this problem based on a single hyperparameter and 8 smaller regression tasks, which is insufficient empirical evidence. In addition, any other hyperparameters of strategies, model architecture, and model training, for instance, were chosen from previous works, which contradicts the notion that good hyperparameters are unknown in real-world applications. Another aspect is the quality of writing, where I see deficiencies in precision and soundness along with ambiguous usage of terms. For example, the terms "DAL," "DAL model," and "DAL method" are used interchangeably.

**Questions:**

I would consider testing different query sizes (k), as this can strongly influence the performance of uncertainty and diversity-based strategies. Roughly said, uncertainty works better for smaller batch sizes, while diversity benefits from larger batch sizes.

Following the first point, starting with 80 labeled samples and selecting an additional 40 labeled samples until reaching 2080 labeled samples is a large pool of labeled datasets. At least considering two different budgets would bring more clarity and stronger empirical evidence.

The storyline (and the title) does not match the content. too much expectations are raised. I would consider re-framing the article as an investigation of the pool ratio for regression problems.

If the article criticizes how others choose their hyperparameters and the main motivation is that good hyperparameters are not known in real-world applications, the assumption that "appropriate neural network architectures are known apriori" can be seen as a significant contradiction. Similarly, the selection process of dataset-dependent hyperparameters for weight decay and learning rate seems problematic in this context.

One point that has been completely missed is that reducing the pool ratio can enforce a certain diversity in DAL strategies, as can be seen in Figure 2, which may explain some of the preferences of smaller pool ratios for uncertainty-based strategies or the negative correlation between pool diversity and pool ratio for BALD and LearningLoss in Figure 5.

I would consider pointing out that a pool ratio cannot be optimized during real-world application - even if a validation dataset exists. This could further motivate the work and set it apart from other work that, for example, optimizes weight decay during the DAL experiment.

---

### Official Review · Reviewer_hdga · 2024-11-04

**Soundness:** 2
**Presentation:** 3
**Contribution:** 2
**Rating:** 5
**Confidence:** 4

**Summary:**

The paper addresses the gap in evaluating the robustness of DAL methods under real-world (or "in the wild") conditions, where the optimal HPs are not known in advance. Many DAL studies assume hyperparameters, particularly the pool ratio ($\gamma$), are pre-optimized. However, in practical applications, this assumption is unrealistic. The paper identifies that the sensitivity to $\gamma$ might make DAL methods perform worse than random sampling in many cases​.

The authors benchmark 11 DAL methods across 8 regression tasks, varying $\gamma$ to evaluate their robustness and performance under real-world uncertainty​. Most DAL methods underperform relative to random sampling when $\gamma$ is uncertain, but diversity-focused methods (GSx, GSxy, QBCDiv) show consistent, robust performance, highlighting the value of diversity in DAL selection criteria.

**Strengths:**

1. This paper studies whether deep active learning algorithms are practical in real-world applications, which is an important but overlooked problem.

2. It benchmarks 11 DAL methods across 8 regression tasks, providing a thorough empirical analysis. The authors’ choice of real-world, emerging scientific datasets like aerodynamics and materials science adds practical value, testing DAL methods in diverse, high-impact applications and enhancing the study's relevance.

3. The evaluation shows that 8 out of 11 methods do not always guarantee to outperform the random sampling with a varying hyperparameter.

**Weaknesses:**

1. The paper only evaluates regression scientific tasks. It will be good to include more complicated vision or NLP tasks to better demonstrate that the conclusion is valid in other scenarios.

2. The entire evaluation is all about a single hyperparameter -- pool ratio, which limits the contribution of the paper. The paper can be more solid if more factors are considered and evaluated.

3. An interesting follow-up problem can be that, if many DAL methods fail to achieve good performance when hyperparameters change, can we design a better method to solve this problem, like automatically choosing the best HP? Such algorithms that can address the problem can also increase the contribution of the paper.

4. It can benefit the paper to have a more detailed table on the 8 regression tasks - dataset size, detection goal, etc.

**Questions:**

Is Figure 2 a conceptual illustration, or is it based on a specific dataset or task? Additionally, what does the range of the x-axis represent?

---

### Official Review · Reviewer_toR1 · 2024-11-04

**Soundness:** 2
**Presentation:** 2
**Contribution:** 2
**Rating:** 3
**Confidence:** 5

**Summary:**

This is a benchmark paper, the author evaluates eleven state-of-the-art DAL methods across eight datasets in the wild.

**Strengths:**

1. It tests eleven different DAL methods across various problems.

2. It highlights that methods using sample diversity are more reliable.

3. The findings might be useful for people who want to use DAL in practical settings.

**Weaknesses:**

1. Setting the pool ratio itself is not meaningful, since we can first use diversity-based measures to determine the subset and then use uncertainty-based measures to get better AL performance.

2. Datasets are too simple.

3. The conclusions of this paper rely on empirical evidence to support their findings. For example, it lacks a formal theoretical framework or mathematical proof to explain why diversity-based methods are inherently more robust.

4. As the author mentioned "The recent study by Holzmüller et al. (2023) is the only work that is similar to ours", however, the comparisons described in this paper are not strong enough to explain the novelty and necessity of this paper.

**Questions:**

1. These figures are really hard to read, like Figure 3, and MSE plots in supplementary materials.

---

### Author Response · Authors · 2024-12-02
**We plan to withdraw - thank you to the reviewers for their time!**

We wish to thank the reviewers for taking the time to review our manuscript - we greatly appreciate it!

While we believe that we can address many of the reviewers' concerns, given the nature of the feedback and the scores, we believe it best to withdraw the paper and revise/resubmit elsewhere.  Thank you all again!

---

### Note · Authors · 2024-12-22

**Comment:**

Confirming that we have decided to withdraw our submission. We thank the reviewers and meta-reviewers for their time.

**Withdrawal Confirmation:**

I have read and agree with the venue's withdrawal policy on behalf of myself and my co-authors.